# Cell-cycle-gated feedback control mediates desensitization to interferon stimulation

Anusorn Mudla[1], Yanfei Jiang[1], Kei-ichiro Arimoto[1], Bingxian Xu[1], Adarsh Rajesh[2], Andy P Ryan[1], Wei Wang[3], Matthew D Daugherty[1], Dong-Er Zhang[1,4], Nan Hao[1]*

[1]Section of Molecular Biology, Division of Biological Sciences, University of California, San Diego, La Jolla, United States; [2]Department of Bioengineering, University of California, San Diego, La Jolla, United States; [3]Department of Chemistry and Biochemistry, University of California, San Diego, La Jolla, United States; [4]Department of Pathology, Moores UCSD Cancer Center, University of California, San Diego, La Jolla, United States

**Abstract** Cells use molecular circuits to interpret and respond to extracellular cues, such as hormones and cytokines, which are often released in a temporally varying fashion. In this study, we combine microfluidics, time-lapse microscopy, and computational modeling to investigate how the type I interferon (IFN)-responsive regulatory network operates in single human cells to process repetitive IFN stimulation. We found that IFN-$\alpha$ pretreatments lead to opposite effects, priming versus desensitization, depending on input durations. These effects are governed by a regulatory network composed of a fast-acting positive feedback loop and a delayed negative feedback loop, mediated by upregulation of ubiquitin-specific peptidase 18 (USP18). We further revealed that USP18 upregulation can only be initiated at the G1/early S phases of cell cycle upon the treatment onset, resulting in heterogeneous and delayed induction kinetics in single cells. This cell cycle gating provides a temporal compartmentalization of feedback loops, enabling duration-dependent desensitization to repetitive stimulations.

*For correspondence: nhao@ucsd.edu

Competing interests: The authors declare that no competing interests exist.

## Introduction

Under physiological conditions, cells often encounter environmental cues that change over time. Many hormones, cytokines, and signal factors are released in a temporally varying fashion. Increasing evidence demonstrated that cells can use complex signaling networks to interpret the dynamic patterns of these inputs and initiate appropriate cellular responses (*Purvis and Lahav, 2013*; *Behar and Hoffmann, 2010*). For example, the mitogen-activated protein kinase Hog1 pathway in the yeast *Saccharomyces cerevisiae* responds to the various frequencies of oscillating osmotic stress and differentially control the growth rate under stress (*Mitchell et al., 2015*; *Hersen et al., 2008*; *Mettetal et al., 2008*). Moreover, the gene regulatory program mediated by the yeast general stress-responsive transcription factors (TFs) Msn2 and Msn4 can decode various input pulses and induce differential gene expression (*Hao and O'Shea, 2012*; *Hao et al., 2013*; *Hansen and O'Shea, 2013*; *AkhavanAghdam et al., 2016*). In mammalian systems, it has been shown that the nuclear factor κB (NFκB) pathway can process the pulsatile stimulation of tumor necrosis factor-α (TNF-α) to determine the timing and specificity of downstream gene expression (*Ashall et al., 2009*; *Tay et al., 2010*; *Nelson et al., 2004*). Similarly, the p53 tumor suppressor differentially regulates target genes and cell fates by processing temporal patterns of DNA damage cues (*Harton et al., 2019*; *Purvis et al., 2012*; *Batchelor et al., 2011*). Intriguingly, many of these studies observed that individual cells exhibit widely different behaviors even to the same stimuli, and, as a result, population-

based measurements may obscure the actual response dynamics of individual cells, leading to inaccurate interpretation of the data. Furthermore, these observed cell-to-cell variabilities play important roles in enhancing the diversity of physiological behaviors and biological functions (*Hsu et al., 2019*; *Reyes et al., 2018*; *Yang et al., 2017*; *Paek et al., 2016*; *Min et al., 2020*). In this study, we focus on interferon (IFN)-α signaling in HeLa cells and investigate how the IFN-driven gene regulatory network operates in single human cells to decode various signal dynamics.

IFN-α is a member of the type I IFN family of cytokines, which are synthesized and secreted in mammals upon pathogen infection and initiate innate immune responses to limit pathogen spread via reducing protein production, upregulating antiproliferative and antiviral genes, and programmed cell death (*Schneider et al., 2014*; *Barber and defense, 2001*). IFN-α has also been clinically used in treatments of a variety of diseases, such as hepatitis B and C infection, HIV infection, melanoma, kidney cancer, leukemia and lymphoma (*Watanabe et al., 2013*; *Medrano et al., 2017*). IFN-α exerts its anti-pathogenic and anti-proliferative effects by activating the Janus kinase (JAK)-signal transducer and activator of transcription (STAT) pathway, leading to the expression of over 300 IFN-stimulated genes (ISGs) (*Schneider et al., 2014*; *Schoggins and Rice, 2011*). IFN-α binds to a heterodimeric transmembrane receptor, the IFN-α receptor (IFNAR), triggering the activation of receptor-associated kinases Janus kinase 1 (JAK1) and tyrosine kinase 2 (TYK2), which in turn phosphorylate transcription factors signal transducer and activator of transcription 1 (STAT1) and STAT2. The phosphorylated STAT1 and STAT2 dimerize and associate with IFN-regulatory factor 9 (IRF9) to form IFN-stimulated gene factor 3 (ISGF3) complex. ISGF3 then translocates to the nucleus and binds to the DNA consensus sequences, known as IFN-stimulated response element (ISRE), activating the transcription of ISGs (*Platanias, 2005*; *Schreiber, 2017*). The duration and strength of the IFN-mediated inflammatory responses are tightly controlled in mammals. A response that is too short or too weak will fail to limit pathogen spread, whereas a response that is too prolonged or too strong will result in tissue damage, organ failure, and carcinogenesis (*Choubey and Moudgil, 2011*; *Crow, 2016*; *McNab et al., 2015*). In many epidemics, uncontrolled inflammatory responses to infection have led to the cytokine storm and high mortality (*Cameron et al., 2008*; *Carrero, 2013*).

Although the molecular components of the JAK-STAT pathway have been well characterized, how they are regulated to generate appropriate responses to dynamic IFN-α inputs remains largely elusive. In particular, during chronic inflammation, cells receive varying IFN signals from neighboring cells (*Beltra and Decaluwe, 2016*; *Landskron et al., 2014*; *Perry et al., 2005*). However, previous studies have reported opposing results regarding the effect of IFN-α pretreatment. In some studies, a prior exposure to IFN-α accelerates cells' responses to the second IFN input, enabling a 'priming' effect (*Abreu et al., 1979*; *Kuri et al., 2009*; *Rodriguez-Pla et al., 2014*; *Phipps-Yonas et al., 2008*). In other studies, however, a pretreatment with IFN-α diminishes the responses to the following stimulation, resulting in a 'desensitization' effect (*Scagnolari and Antonelli, 2018*; *Sarasin-Filipowicz et al., 2009*; *Makowska et al., 2011*). To resolve this paradox, we combined time-lapse imaging, microfluidics, and computational modeling to track and quantify JAK-STAT signaling and downstream gene expression in single human epithelial cells. We revealed that a cell cycle-gated negative feedback loop functions to decode IFN pretreatments with different durations and induce differential single-cell responses, reconciling the opposing results from previous studies. Our findings unravel the important role of the dynamics of IFN pretreatments on modulating cellular responses, suggesting that optimizing administration timing may help boosting the effectiveness of IFN treatment.

## Results

### IFN-α pretreatments confer opposite effects depending on their durations

IFN-α activates the JAK-STAT pathway and upregulates the expression of IFN-stimulated genes (ISGs), initiating an acute inflammatory response to limit pathogen spread. Recent studies found that individual cells respond to IFNs in a highly heterogeneous manner, both in vitro (*Czerkies et al., 2018*; *Rand et al., 2012*; *Patil et al., 2015*) and in vivo (*Miao et al., 2010*; *Stifter et al., 2019*). Therefore, traditional assays that measure averaged responses across populations at static time points may not be able to accurately characterize cells' responses to IFNs. To monitor the dynamics

of JAK-STAT signaling at the single-cell level, we constructed a reporter cell line in HeLa cells, in which *STAT1* was C-terminally tagged with mCherry at its native locus using CRISPR/Cas9. In addition, to monitor downstream gene expression, we inserted a yellow fluorescent protein (YFP) under the endogenous promoter of a representative ISG *IRF9* (P$_{IRF9}$), with a translational skip spacer (P2A) between the reporter and the *IRF9* coding region (*Figure 1A*; *Figure 1—figure supplement 1, A–C*). We chose *IRF9* because it has been well defined as one of the major early response ISGs and it encodes a key component of the ISGF3 complex (*Cheon et al., 2013*; *Tsuno et al., 2009*). Using this reporter cell line and time-lapse microscopy, we were able to simultaneously track nuclear translocation of STAT1 and downstream *IRF9* expression in a large number of single cells (*Figure 1B*). We observed a rapid nuclear translocation (within ~0.5 hr) of STAT1 followed by a gradual increase in P$_{IRF9}$-driven gene expression in response to IFN-α stimulation (*Figure 1B and C*; *Video 1*), consistent with traditional western blotting results (*Figure 1—figure supplement 1, D*). For each single cell, we quantified the nuclear to cytoplasmic ratio of STAT1, which resembles STAT1 activation, and the rate of increase in YFP fluorescence (dYFP/dt) to reflect the P$_{IRF9}$-driven transcriptional activity. STAT1 nuclear translocation correlated temporally with the increase in P$_{IRF9}$-driven transcriptional activity (*Figure 1C*). We examined single-cell responses to various doses of IFN-α (*Figure 1—figure supplement 2*) and chose 100 ng/ml (10,000 IU/ml), a sub-saturating and clinically relevant (*Tarhini et al., 2012*) concentration, for the following analyses.

To examine the responses to repetitive IFN-α stimulation, we employed a previously reported microfluidic device designed for mammalian cell culturing (*Kolnik et al., 2012*). The device features rapid cell loading by on-chip vacuum and long-term cell culturing in chambers that are isolated from shear stress. In this study, we modified the device to enable constant flows for medium refreshing and computer-programmed dynamic control of IFN inputs (*Figure 1D*). Using this device, we first exposed the cells to a pulse of IFN-α pretreatment with different durations, followed by an 8-hr break time with the normal medium. We then imposed a second 10-hr IFN-α treatment and evaluated single-cell responses. The device was integrated with time-lapse microscopy to allow tracking of a large number of single cells throughout the experiments (*Figure 1D*). We found that a 2- or 10-hr pretreatment accelerated and enhanced the transcriptional response to the second IFN-α input, as indicated by increased induction rates and levels of the P$_{IRF9}$-driven reporter, compared to the control without pretreatment (*Figure 1E and F*; compare green and blue curves/bars with black ones). Intriguingly, a 24-hr pretreatment, however, dramatically decreased the transcriptional response to the second input (*Figure 1E and F* compare red curves/bars with black ones). Therefore, a pretreatment of IFN-α, depending on its duration, could lead to opposite effects on the responses to the subsequent input. A short pretreatment induces a priming effect, whereas a prolonged pretreatment triggers desensitization. We also examined how changing the dose of IFN pretreatment impacts the effect on the response to the second input and found that a dose higher than 10 ng/ml is required for desensitization in 24-hr pretreatment (*Figure 1—figure supplement 3, A–C*). Furthermore, we tested how the break time between pretreatment and second stimulation influences desensitization and found that the effect of desensitization decreased as a function of break time (*Figure 1—figure supplement 4*).

## USP18 is responsible for desensitization induced by the prolonged IFN-α pretreatment

We next considered the mechanisms underlying the opposite effects induced by IFN pretreatments. Previous studies have demonstrated that the priming effect can be attributed to the expression induction of IRF9, STAT1 and STAT2, which are components of ISGF3 transcriptional complex that mediates IFN-driven gene expression (*Cheon et al., 2013*), and IFN-induced chromatin modifications (*Kamada et al., 2018*). To determine the mechanism of desensitization caused by the prolonged pretreatment, we use a short hairpin RNA (shRNA) to knock-down the expression of ubiquitin-specific peptidase 18 (USP18), a major negative regulator of JAK-STAT signaling that we identified previously (*Zou et al., 2007*; *Kim et al., 2008*; *Ritchie et al., 2004*; *Malakhova et al., 2003*), in the reporter cell line (*Figure 2—figure supplement 1*, USP18-KD). USP18 is transcriptionally upregulated by IFN treatment and exerts inhibition of IFN-α signaling at the receptor level, forming a negative feedback loop (*Sarasin-Filipowicz et al., 2009*; *François-Newton et al., 2011*; *Arimoto et al., 2017*; *Malakhova et al., 2006*).

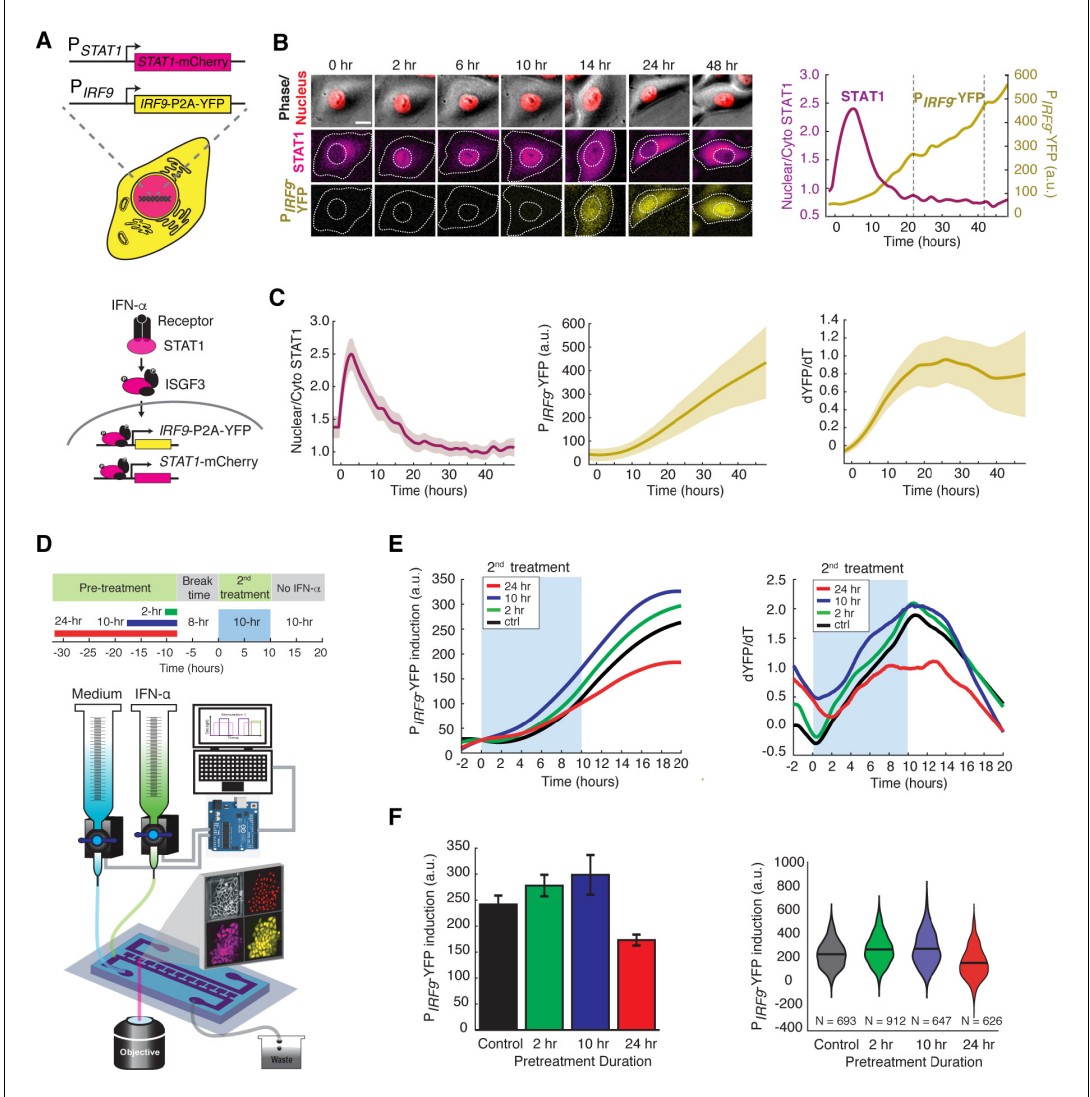

**Figure 1.** IFN-α pretreatments with different durations lead to opposite effects to the second stimulation. (**A**) Schematic of HeLa reporter cell line engineered using CRISPR/Cas9 (top). *STAT1* was tagged with mCherry at C-terminus to monitor the translocation and expression. The coding sequence for P2A-YFP was inserted at the C-terminus of *IRF9* coding sequence to generate a transcription reporter (P_IRF9). A simplified diagram of the IFN-α pathway components that can be monitored using the reporter cell line (bottom). (**B**) Time lapse images of a representative cell treated with 100 ng/ml IFN-α for 48 hrs. Scale bar: 20 μm. The time traces of nuclear/cytoplasmic STAT1-mCherry and P_IRF9-YFP signals of the cell are shown on the right. Vertical dashed lines represent cell divisions. (**C**) Averaged time traces of nuclear/cytoplasmic STAT1-mCherry, P_IRF9-YFP, and the time derivative of P_IRF9-YFP (dYFP/dt) (n = 257 cells). Data are represented as the mean (solid lines) and $\pm$ standard deviation (SD) (shaded areas). (**D**) Schematic of IFN-α pretreatment experiments (top). Cells were pretreated with 100 ng/ml IFN-α for 0, 2, 10 and 24 hrs followed by 8 hrs of break time and re-stimulated with 100 ng/ml IFN-α for an additional 10 hrs. Bottom: A diagram of the microfluidic set-up. Two syringes filled with culture medium with or without IFN-α were connected to programmable Arduino-controlled valves that control the duration of IFN-α treatments. Images were captured every 5 min throughout the entire experiment that lasted for a total of 52 hr. (**E**) Averaged time traces of P_IRF9-driven YFP induction (left) and the rate of induction (dYFP/dt, right) in response to the second IFN-α treatment under different pretreatment conditions. For P_IRF9-YFP induction, the baselines at the beginning of the second stimulation were normalized to the same level for the comparison of induction levels under different pretreatment conditions. Results were from at least three independent experiments. (**F**) Amounts of P_IRF9-YFP induction by the second IFN-α stimulation under different pretreatment conditions were shown in bar graph (left) and violin plot (right). The bar showed the averages from three independent experiments, represented as mean $\pm$ standard error of the mean (SEM). The differences between the pretreatment conditions and the control are all statistically significant (p<0.001). The violin plots showed the distributions of single-cell responses under different pretreatment conditions. The violin plots cover single-cell data from three independent experiments.

The online version of this article includes the following figure supplement(s) for figure 1:

**Figure supplement 1.** Cell line construction and validation.

**Figure supplement 2.** Dose-dependent responses to IFN-α treatment.

*Figure 1 continued on next page*

*Figure 1 continued*

**Figure supplement 3.** Dose dependence of desensitization to IFN-α treatment.
**Figure supplement 4.** Dependence of desensitization effects on the break time.

We found that a 24-hr IFN-α pretreatment substantially diminished STAT1 nuclear translocation in WT cells upon the second IFN input, whereas 2-hr or 10-hr pretreatment shows a modest effect (*Figure 2A*, 'WT'; *Figure 2—figure supplement 2, A and B*). However, this desensitization effect was abolished when USP18 expression was knocked down (*Figure 2A*, '*USP18-KD*'; *Figure 2—figure supplement 2, C and D*). Furthermore, in accord with STAT1 nuclear translocation, IFN-α pretreatments boosted the transcriptional responses in *USP18*-KD cells upon the second input, exhibiting priming effects independent of their durations or doses (*Figure 2*, B and C; *Figure 1—figure supplement 3, D–F*). These results indicate that USP18 is required for desensitization. We also examined the role of suppressors of cytokine signaling 1 (SOCS1), another negative regulator of JAK-STAT signaling (*Liau et al., 2018*). In contrast to *USP18-KD*, knocking down *SOCS1* (*SOCS1-KD*) did not affect the desensitization of STAT1 nuclear translocation by the prolonged IFN pretreatment (*Figure 2—figure supplement 3*). Moreover, to determine how USP18-mediated desensitization influences the antiviral effect of IFN-α treatment, we examined viral replication in WT and *USP18-KD* with or without 24-hr IFN-α pretreatment. We found that the IFN-α pretreatment resulted in a much more dramatic repression of viral replication in *USP18-KD*, ~15 times more effective than in WT (*Figure 2—figure supplement 4*), indicating that USP18-mediated desensitization attenuates the antiviral effect of prolonged IFN-α pretreatment.

## Computational modeling suggests a delayed negative feedback loop through USP18

Based on our experimental results, we postulated that the opposite effects induced by short versus prolonged pretreatment inputs might be caused by different expression kinetics of ISGF3 components and USP18: a short input is sufficient to trigger ISGF3 expression and thereby the priming effect, whereas a prolonged input is required to induce USP18 expression and hence desensitization. To test this hypothesis in silico, we devised a simple computational model, which is composed of two ordinary differential equations that govern the expression of *IRF9*, an ISGF3 component, and *USP18*. In this model, IRF9 and USP18 act as the positive and the negative feedback regulators of the JAK-STAT pathway, respectively. Another major difference is that, as we proposed above, the upregulation of *USP18* expression features a delayed kinetics and hence requires a continuous IFN stimulation that lasts longer than the delay time τ, whereas the upregulation of *IRF9* initiates immediately upon the IFN treatment (*Figure 3A* and *Figure 3—figure supplement 1A*; see Materials and methods for details).

This model was able to reproduce the results from pretreatment experiments, where the priming effect dominates for the short pretreatments (2- and 10-hr pretreatments) and the desensitization effect dominates when the pretreatment duration increases to 24-hr (*Figure 3C*; the data from *Figure 1F*), and positive and negative feedback loops are both important for these effects (parameter sensitivity analysis and model comparison are shown in *Figure 3—figure supplement 1, D and E*). To investigate the effect of the delay in *USP18* upregulation, we altered the delay time from 1 to 20-hr while keeping all the other parameters free. By fitting the model to the data for each assigned delay time, we found that the fitting error (between simulations and data) reaches the minimum when the delay time is 8 hrs (*Figure 3*, B and C; *Figure 3—figure supplement 1, B–E*). These analyses supported

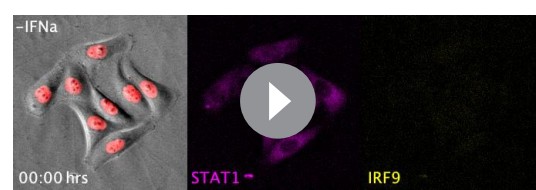

**Video 1.** Response of the dual reporter cell line to 100 ng/ml IFN-α treatment. The image sequence was acquired for 50 hrs in which the first 2 hr were before the addition of IFN-α. The NLS-2xiRFP nuclear marker is shown in red and is merged with phase images of the cells. STAT1-mCherry and P$_{IRF9}$-YFP in the same cells are also shown. Movie is shown at 50 frames/second.

https://elifesciences.org/articles/58825#video1

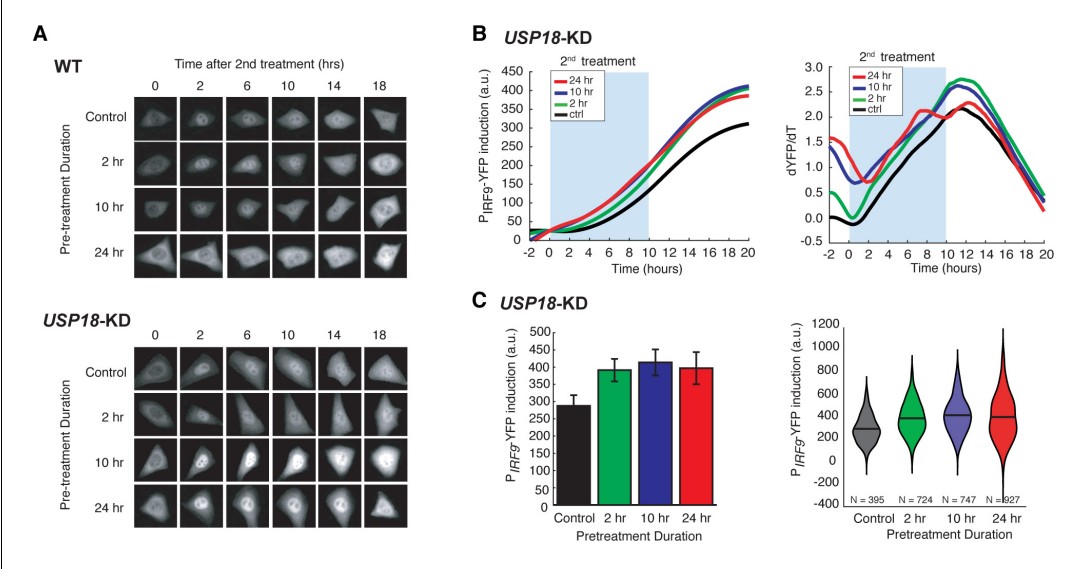

**Figure 2.** USP18 mediates desensitization induced by the prolonged IFN-α pretreatment. (**A**) Representative time-lapse images of STAT1 nuclear translocation in response to the second IFN-α treatment under different pretreatment conditions in WT cells (top) and *USP18*-KD (bottom). (**B**) Averaged time traces of P$_{IRF9}$-driven YFP induction (left) and the rate of induction (dYFP/dt, right) in *USP18*-KD cells in response to the second IFN-α treatment under different pretreatment conditions. (**C**) Amounts of P$_{IRF9}$-YFP induction in *USP18*-KD cells by the second IFN-α stimulation under different pretreatment conditions were shown in bar graph (left) and violin plot (right). The bar showed the averages from three independent experiments, represented as mean ± standard error of the mean (SEM). The differences between the pretreatment conditions and the control are all statistically significant (p<0.001). The violin plots showed the distributions of single-cell responses under different pretreatment conditions. The violin plots cover single-cell data from three independent experiments.

The online version of this article includes the following figure supplement(s) for figure 2:

**Figure supplement 1.** Validation of the *USP18*-KD cell line.

**Figure supplement 2.** Quantification of USP18-mediated desensitization in STAT1 nuclear translocation.

**Figure supplement 3.** SOCS1 does not mediate the desensitization of STAT1 nuclear translocation upon IFN stimulation.

**Figure supplement 4.** Vesicular stomatitis virus (VSV) replication in WT and *USP18*-KD cells with and without 24-hr IFN-α pretreatment.

the hypothesis about a substantial delay in *USP18* upregulation. The model further predicted that repetitive transient IFN inputs that are shorter than the delay time could be less able to induce the USP18-mediated negative feedback loop and hence could lead to a higher transcriptional response than that induced by a prolonged input with the same total treatment time. For example, pulsatile inputs with 5 × 8-hr pulses could produce a higher transcriptional response than that induced by a 40-hr sustained input, and the difference should be USP18-dependent (*Figure 3*, D and E). This prediction was tested experimentally and a higher transcriptional response, indicated by the P$_{IRF9}$-driven reporter, was observed when 5 × 8-hr IFN pulses were given, compared to that induced by a single 40-hr treatment. Furthermore, this difference caused by different input dynamics was abolished in *USP18-KD* cells (*Figure 3F*; *Figure 3—figure supplement 2*).

In summary, our modeling results suggest that a prolonged input is required to initiate *USP18* upregulation. Therefore, when the input duration is short, the ISGF3-mediated-positive feedback loop dominates the regulation of JAK-STAT pathway, conferring the priming effect to subsequent stimulation. Once the input duration is prolonged enough to induce USP18 upregulation, the negative regulation by USP18 overrides the positive regulation by ISGF3, resulting in desensitization.

## The kinetics of *USP18* upregulation by IFN is heterogeneous in single cells

To test our model directly, we set out to compare the expression kinetics of *IRF9* and *USP18* in living cells. To this end, we built upon the dual reporter cell line constructed in *Figure 1A* and inserted a cyan fluorescent protein (CFP) with a nuclear localization signal (NLS) under the endogenous *USP18* promoter (P$_{USP18}$) with a P2A spacer between the reporter and the *USP18* coding region

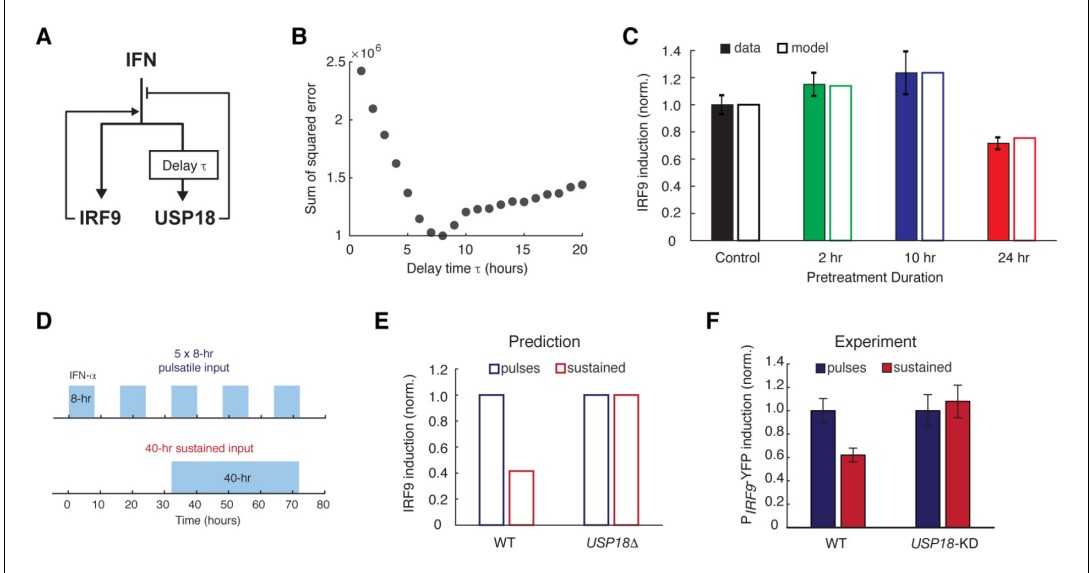

**Figure 3.** A kinetic model suggests a delayed negative feedback loop through USP18. (**A**) A diagram for the simple kinetic model of the IFN-driven gene regulatory network. (**B**) Fitting errors between simulations and the data with different assigned values of the delay time in USP18 upregulation. (**C**) Amounts of $P_{IRF9}$-YFP induction by the second IFN-α stimulation under different pretreatment conditions, from experimental data (solid bars) and model simulations with the best-fit parameters (open bars). Data were from *Figure 1F* and were normalized to the non-pretreatment condition (control). (**D**) Schematic of experimental design with repetitive IFN pulses versus a sustained IFN input. (**E**) Model prediction of the responses to pulsatile versus sustained IFN inputs in the presence and absence of USP18. Results were normalized to the amount of induction to the pulsatile IFN input in WT. (**F**) Experimental data of the responses to pulsatile versus sustained IFN inputs in WT and *USP18*-KD cells. The error bars represent standard deviations of single-cell data.

The online version of this article includes the following figure supplement(s) for figure 3:

**Figure supplement 1.** Model fitting results and the parameter analysis.

**Figure supplement 2.** Pulsatile IFN-α treatment induces higher ISG expression in single cells.

(*Figure 4A*; *Figure 4—figure supplement 1*). This cell line enabled us to simultaneously track the kinetics of $P_{IRF9}$ and $P_{USP18}$-driven gene expression in the same cells.

To evaluate the temporal difference in *IRF9* and *USP18* upregulation, we measured and quantified, in each individual cell, the induction kinetics of $P_{IRF9}$ and $P_{USP18}$ -driven gene expression upon IFN stimulation. We defined the time needed for induction to initiate as 'activation time' and the difference between the activation times of *IRF9* and *USP18* as 'delay time' (*Figure 4B*). We found that, consistent with our model, *IRF9* and *USP18*, although induced by the same upstream JAK-STAT signaling, exhibited strikingly different activation times. $P_{IRF9}$-driven expression was induced with a fast and relatively uniform kinetics among cells; in contrast, $P_{USP18}$-driven expression exhibited a slow and heterogeneous kinetics (*Figure 4C*). Intriguingly, we quantified the delay times of *UPS18* induction, relative to that of *IRF9*, in single cells and observed a long-tail distribution: about 83.3% of cells showed modest delays in induction (<10 hrs), whereas the other 16.7% cells exhibited more prolonged delays (>10 hrs). We classified these two subpopulations as 'Group 1' and 'Group 2', respectively (*Figure 4D*).

To determine the source of clonal heterogeneity in *USP18* induction, we analyzed the relationship of USP18 expression with STAT1 or IRF9 expression at the single-cell level and found that the heterogeneity in these upstream components contributed partially to cell-to-cell variation in the expression levels of USP18. However, the delay times of *USP18* upregulation correlated poorly with STAT1 or IRF9 expression (*Figure 4—figure supplement 2*). We further considered the contribution of the cell cycle stage at IFN treatment onset, as the cell cycle progression has been shown as a major factor that coordinates gene expression (*Liu et al., 2017*). To this end, we quantified the extent of cell cycle progression in each single cell at IFN treatment onset ('% of cell cycle progression'; *Figure 4—figure supplement 3*) and examined its relationship with the *USP18* delay time in the same cell. We found that cells showed modest delays if the treatment was added at the early phase of their cell

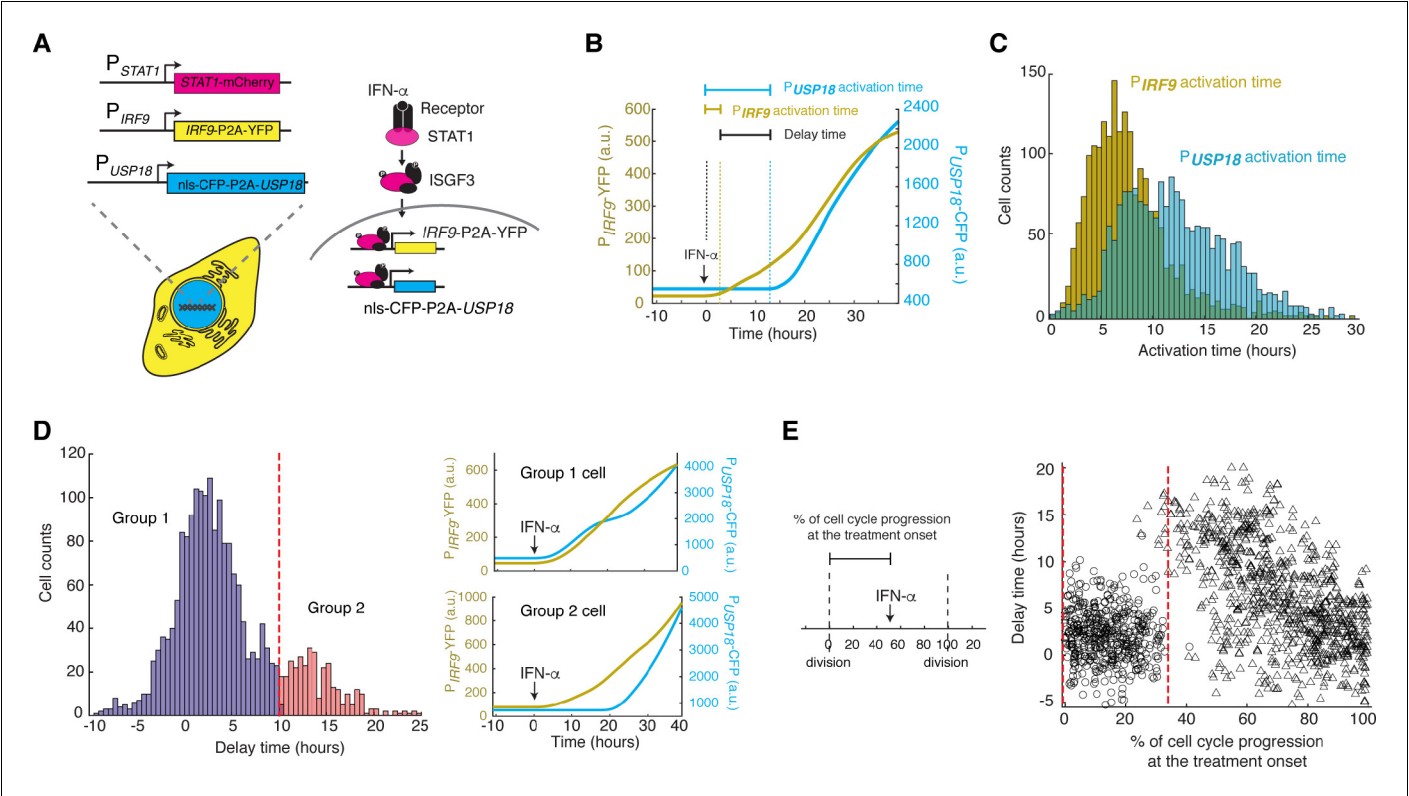

**Figure 4.** Heterogeneous delays in *USP18* upregulation by IFN were observed in single cells. (**A**) Schematic of the dual reporter cell line. A coding sequence for NLS-CFP-P2A was inserted endogenously into the N-terminus of *USP18* coding sequence of the previous cell line. IFN-α induces nuclear translocation of STAT1 and upregulation of *IRF9* (P$_{IRF9}$-YFP) and *USP18* (P$_{USP18}$-CFP). (**B**) Representative time traces of P$_{IRF9}$-YFP and P$_{USP18}$-CFP of a single cell in response to IFN-α. Activation time is defined as the time required to initiate the upregulation of the reporters after IFN-α treatment onset. Delay time is defined as the time difference between P$_{USP18}$ and P$_{IRF9}$ activation times. (**C**) Distributions of P$_{IRF9}$ and P$_{USP18}$ activation times in single cells (n = 2021 cells). The mean activation times (with 95% confidence interval) of *IRF9* and *USP18* are 7.9008 ± 0.1914 and 12.1620 ± 0.2349 hrs, respectively. The coefficients of variation (CVs) for activation times of *IRF9* and *USP18* are 0.5558 and 0.4429, respectively. (**D**) Distributions of delay times in single cells, quantified from the activation times in C. Cells are classified into two groups based on the delay times. Representative time traces of P$_{IRF9}$ and P$_{USP18}$ in a single cell from each group are shown (right). Proportion of Group 2 cells (with a delay longer than 10 hrs) is 16.48%. (**E**) Single-cell delay times as a function of the percentages of cell cycle progression upon IFN treatment onset. Left: Diagram illustrating the quantification of the percentage of cell cycle progression at the treatment onset in a single cell. The time between two cell divisions (dashed lines) was considered as the length of one cell cycle. % of cell cycle progression is calculated as the ratio of the time in a cell cycle before IFN-α addition versus the full cell cycle length (100%) (See *Figure 4—figure supplement 3* for details). Right: Scatterplot of delay time in each single cell versus % of cell cycle progression upon treatment onset. Open circles represent cells in which P$_{USP18}$-CFP upregulation occurred within the same cell cycle as the IFN-α addition. Open triangles represent cells in which P$_{USP18}$-CFP upregulation occurred in the next cell cycle. Red dashed lines indicate the time window for immediate P$_{USP18}$ induction. A two-sample t-test was performed for the two populations within and beyond the time window and obtained a p-value<0.001, indicating a significant difference in delay times.

The online version of this article includes the following figure supplement(s) for figure 4:

**Figure supplement 1.** Construction of the cell line with P$_{USP18}$-CFP reporter.

**Figure supplement 2.** Relationships between variations in USP18 upregulation and the heterogeneity in ISGF3 components.

**Figure supplement 3.** Quantifying the percentage of cell cycle progression upon IFN treatment onset in single cells.

cycles (0–35% of the cell cycle). In contrast, in those cells that progressed beyond this phase, the IFN treatment could not initiate USP18 induction until the next cell cycle, resulting in extended delay times proportional to the times needed to reach the next cell cycle (*Figure 4E*). These results suggest that the cell-to-cell variability in USP18 delay times may stem from different cell cycle stages among cells at the treatment onset.

## Cell cycle phases differentially regulate *USP18* expression

Our results in *Figure 4E* suggest that the early phase of cell cycle (0–35% of the cell cycle time), likely the G1 and early S stages, may provide a time window that allows immediate *USP18* induction without significant delays. To directly test that, we imposed chemical perturbations to arrest cells in G1 or G1/S stage and monitored the delay times of *USP18* induction, relative to that of *IRF9*. Specifically, we treated cells with serum starvation (*Langan and Chou, 2011*), lovastatin (*Keyomarsi et al., 1991*; *Javanmoghadam-Kamrani and Keyomarsi, 2008*), and roscovitine (*Kolodziej et al., 2015*), all of which arrested cells in G1 or G1/S stage, prior to the IFN-α treatment. As expected, we observed that cell cycle synchronization substantially reduced the fraction of cells with prolonged delays of *USP18* induction (*Figure 5A*). We confirmed that these decreases in the delay times were primarily due to earlier *USP18* upregulation (*Figure 5—figure supplement 1*).

Furthermore, to explicitly determine the cell cycle stages of individual cells at the treatment onset and their relationships with *USP18* induction delays, we generated another reporter cell line in which a cyclin-dependent kinase 2 (CDK2) activity reporter (*Spencer et al., 2013*) is stably integrated, allowing us to infer the cell cycle stages based on the dynamics of DNA helicase B (DHB) nuclear translocation (*Figure 5B*). In the same cell line, P$_{USP18}$-driven fluorescent reporter was introduced as described in *Figure 4A*. Using this cell line, we classified cells into three groups, G1, S, and G2 cells, based on their cell cycle stages at the IFN treatment onset (*Figure 5C*, left). We found that G1 and S cells showed relatively fast *USP18* activation times, whereas a large fraction of G2 cells exhibited a substantial delay in *USP18* induction (*Figure 5C*, right). We also employed another cell cycle reporter system called fluorescent ubiquitination-based cell cycle indicator (FUCCI) (*Bajar et al., 2016*) and observed similar patterns to those obtained using the CDK2 activity reporter (*Figure 5—figure supplement 2*). These results, together with the cell cycle inhibitor data in *Figure 5A*, confirmed the influence of cell cycle stages on *USP18* induction. The G1/S stages allow rapid induction of expression induction, but the G2 stage restrains the initiation of gene induction, resulting in a prolonged delay in USP18 expression.

What is the molecular mechanism underlying the effects of cell cycle stages on *USP18* expression? Previous studies revealed global variations of epigenetic modifications during cell cycle. For instance, DNA methylation decreases in G1 and increases during S and G2 phases (*Brown et al., 2007*; *Woodcock et al., 1986*; *Desjobert et al., 2015*). We postulated that cell cycle progression might influence USP18 expression through variations in DNA methylation. To test this possibility, we first searched the genome-wide DNA methylation profiles of various human cell types. Intriguingly, the promoter of *USP18* contains a large number of CpG sites, which are highly methylated in all the cell types examined. In contrast, the *IRF9* promoter contains less CpG sites and is less methylated (*Komaki, 2018*; *Figure 5—figure supplement 3*). To determine whether the promoter methylation influences *USP18* expression, we treated cells with decitabine (also known as 5-Aza-2'deoxycytidine), a commonly-used DNA methyltransferase inhibitor (*Christman, 2002*; *Momparler, 2005*). We observed that the decitabine treatment reduced the fraction of cells with prolonged delay times in *USP18* induction, which is in part due to earlier *USP18* upregulation (*Figure 5—figure supplement 4*), supporting that the promoter methylation inhibits *USP18* induction (*Figure 5D*). This result is consistent with a previous study showing that a decreased promoter methylation leads to increased *USP18* expression in breast cancer (*Tan et al., 2018*). We note that decitabine also leads to cell cycle arrest in G1 or G2 (*Shin et al., 2013*; *Lavelle et al., 2003*). Because only a small fraction of decitabine-treated cells is arrested in G1 (*Figure 5—figure supplement 5*) and *USP18* induction is largely delayed in G2 (*Figure 5C*), the accelerating effect of decitabine on *USP18* expression should be via inhibition of methylation rather than cell cycle arrest. In summary, our results suggested that the cell cycle stages may impact the initiation of USP18 upregulation through modulating the promoter methylation level.

## Cell-cycle-gated feedback control shapes single-cell responses to repetitive IFN inputs

Finally, we incorporated the cell-cycle-gated regulation into our model, in replace of the arbitrary delay time, to test whether it is sufficient to account for the experimentally observed delays in *USP18* induction and, consequently, desensitization, at the single-cell level. Based on our experimental results (*Figures 4* and *5*), we assumed that *USP18* can be induced immediately if the IFN input

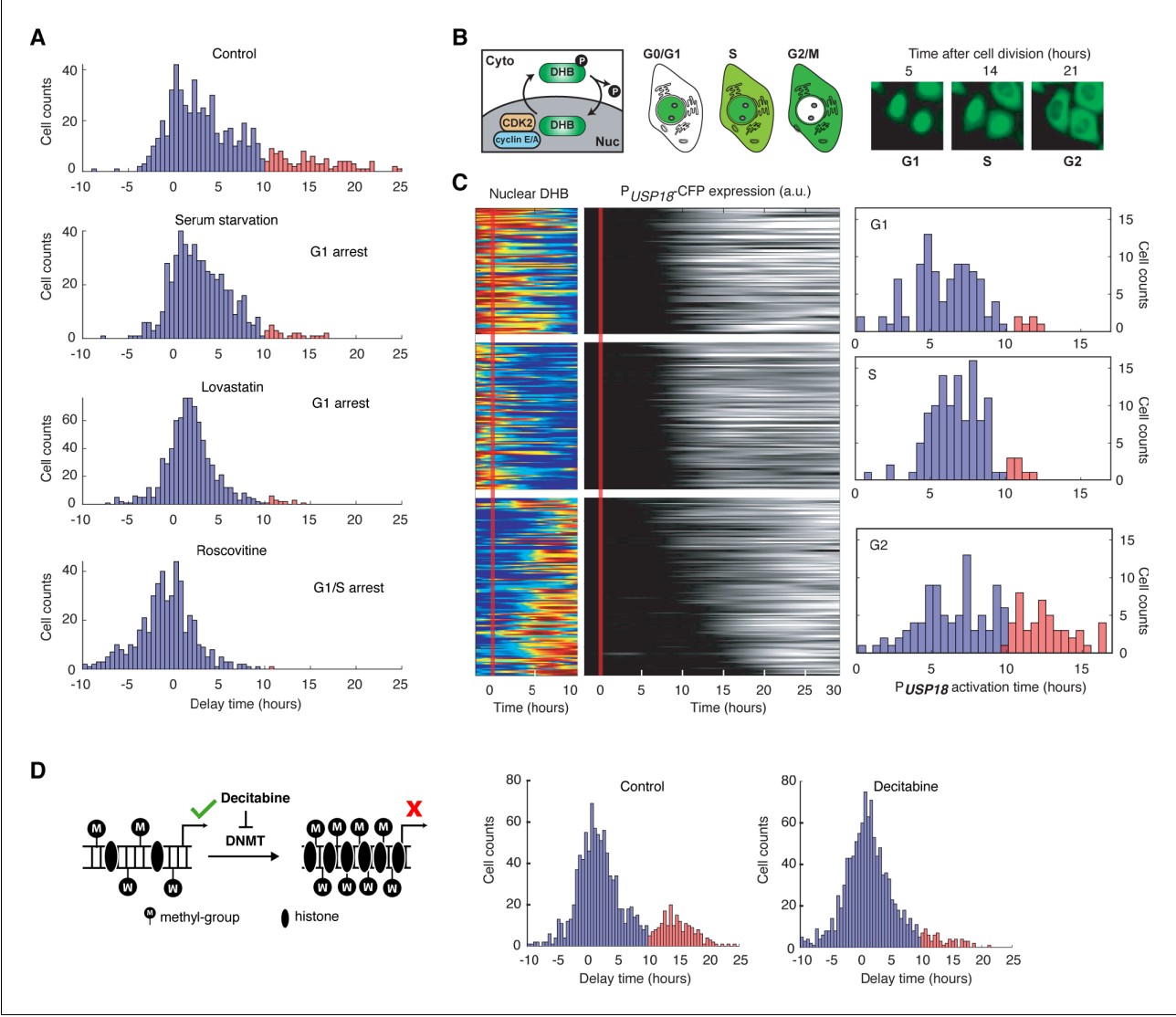

**Figure 5.** *USP18* expression was differentially regulated by cell cycle phases. (**A**) Distributions of delay times in cells treated with different cell cycle perturbations. Cells were serum-starved or treated with lovastatin (5 µM), or with roscovitine (5 µM) for 36 hrs prior to IFN-α treatment. Proportions of Group 2 cells (with a delay longer than 10 hrs) are 16.28% (control), 4.55% (serum starvation), 2.31% (lovastatin), and 0.21% (roscovitine), respectively. (**B**) Schematic of CDK2 activity reporter. Amino acids 994–1087 of human DNA helicase B (DHB) was fused with mCherry. The construct was stably integrated into P$_{USP18}$ cell line using lentivirus. The dynamics of nuclear translocation of DHB-mCherry can be used to infer the cell cycle phase. Representative time-lapse images of DHB-mCherry illustrate the inference of cell cycle phases. (**C**) Color maps showing nuclear DHB and P$_{USP18}$-driven gene expression in the same single cells. Each row represents the time trace of a single cell. Cells were grouped into G1 (n = 104), S (n = 124) and G2 (n = 144) based on the nuclear DHB signals (left) at the time of IFN-α addition. For each group, cells were sorted based on P$_{USP18}$-CFP activation time (middle). Right: Distributions of P$_{USP18}$-CFP activation times for each group. (**D**) Distributions of delay times in cells treated with decitabine, a DNA methyltransferase (DMNT) inhibitor. Left: Schematic of the effect of decitabine on DNA methylation and nucleosome occupancy. Right: Distribution of delay times upon decitabine treatment. Cells were cultured with medium in the absence (control) or presence of 100 µM decitabine for 48 hrs prior to 100 ng/ml IFN-α treatment. Cells with delay times longer than 10 hrs are shown in red. Proportions of Group 2 cells (with a delay longer than 10 hrs) are 17.89% (control) and 5.92% (decitabine treated).

The online version of this article includes the following figure supplement(s) for figure 5:

**Figure supplement 1.** Distributions of P$_{IRF9}$ and P$_{USP18}$ activation times in cells treated with different cell cycle perturbations.
**Figure supplement 2.** Cell cycle-dependent *USP18* upregulation determined by the FUCCI reporter.
**Figure supplement 3.** ISG promoters contain a wide range of CpG site numbers and methylation levels.
**Figure supplement 4.** Distributions of P$_{IRF9}$ and P$_{USP18}$ activation times in cells treated with decitabine.
**Figure supplement 5.** The effect of decitabine on cell cycle.

starts within a fixed time window (G1 and early S phases) of a cell cycle. However, if the input starts outside of the window, cells will have to wait until the open window of the next cell cycle with the delay time equals to the waiting time. We also assumed that the IFN input onset time for each single cell is a random number uniformly distributed within a cell cycle. The length of a full cell cycle was estimated to be 21.82 hrs based on the single-cell time trace measurements (*Figure 2—figure supplement 1*). The model was fit to the experimentally-determined distribution of USP18 delay times within a cell cycle (*Figure 6B*; the data from *Figure 4E*) and we obtained an estimate of the open window length to be 7.3 hrs (~33% of 21.82 hrs). Using these experimentally constrained parameters, we then performed stochastic simulations of single-cell responses to different pretreatment experiments, by introducing noise to gene expression reactions. We observed that the 24-hr pretreatment effectively induced *UPS18* upregulation, to a much higher extent than those upon 2-hr and 10-hr pretreatments. Importantly, as shown in *Figure 6C*, the higher levels of USP18 expression by the prolonged pretreatment lead to reduced IRF9 induction upon the second stimulation at the single-cell level, qualitatively in agreement with our experimental data (*Figure 6D*).

Taken together, our experimental and modeling results suggest that the cell cycle gating can give rise to the delay in *USP18* upregulation and USP18-mediated negative feedback loop, wherein the delay time is largely determined by the length of the open window in the cell cycle (a longer open window results in a shorter delay time). When the duration of pretreatment is shorter than or close to the delay time (e.g. 2- or 10-hr pretreatment), *USP18* can only be partially induced in some cells so that the fast-acting positive feedback loop and the priming effect dominate. In contrast, the 24-hr pretreatment, longer than the entire length of a cell cycle, can fully induce USP18-mediated negative feedback in most cells regardless of their cell cycle stages at treatment onset, resulting in desensitization to subsequent stimulation. In this way, key regulatory processes can be compartmentalized temporally by the cell cycle to decode dynamically varying signals.

## Discussion

IFN signaling is vital in initiating the innate immune response and providing the first line of cellular defense against infection. Although much progress has been made in identifying molecular components that mediate the IFN responses, what remains missing is an understanding about how these components interact and operate dynamically to process varying signals and fine-tune the extent and duration of responses. For example, how cells respond to repetitive IFN stimulation remains puzzling, as previous studies led to opposing conclusions (*Kuri et al., 2009*; *Rodriguez-Pla et al., 2014*; *Sarasin-Filipowicz et al., 2009*; *Makowska et al., 2011*). In this study, we used microfluidics

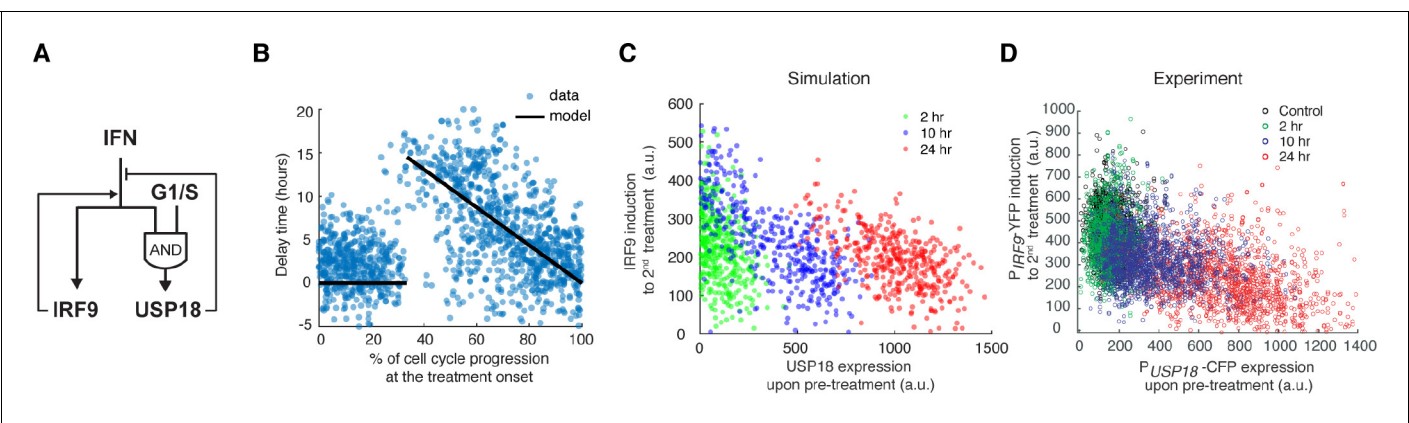

**Figure 6.** Stochastic simulations with the cell-cycle gated feedback control reproduced single-cell responses to IFN pretreatments with different durations. (A) A diagram for the simple model of the IFN-driven gene regulatory network that incorporates the cell cycle gating of USP18 upregulation. (B) Cell-cycle-dependent delay times in the model determined by experimental data from *Figure 4E*. (C) Scatterplot showing the simulated responses under different pretreatment conditions. Stochastic simulations were performed for 400 times for each condition. In the scatterplot, each circle represents a single run with *IRF9* induction in response to the second IFN-α treatment versus *USP18* expression by the pretreatment. (D) Scatterplot showing the single-cell responses under different pretreatment conditions, from experimental data. $P_{USP18}$-CFP expression was measured at the end of the breaktime and the induction of $P_{IRF9}$-YFP was measured 34 hrs after the second IFN-α input was added.

and time-lapse imaging to track the responses to repetitive IFN inputs in single human epithelial cells. We found that the effects of IFN pretreatments are governed by a gene regulatory network comprised of a fast-acting positive feedback loop, in part through upregulation of ISGF3 components, and a delayed negative feedback loop mediated by upregulation of USP18. A short pretreatment input can only induce the positive feedback loop, leading to the priming effect , whereas a prolonged pretreatment activates the negative feedback loop that desensitizes the pathway. Our study reconciled the opposing results from previous studies and revealed that the effects of IFN pretreatments depend on their input durations. This dynamics-based regulation stems from coupled feedback loops that act with different kinetics. Regulatory circuits with coupled negative and positive feedback loops have been discovered in many biological processes, and, depending on the specific mechanisms and kinetics, can give rise to various dynamic behaviors, such as sustained oscillations (*Danino et al., 2010*; *Jiang et al., 2017*) and excitable gene expression (*Ori et al., 2018*; *Süel et al., 2006*; *Mönke et al., 2017*). In this study, the coupled feedback system in the JAK-STAT pathway enables a new function of this well-studied network architecture: it serves as a timer that measures the initial input duration and coordinate the temporal order of regulatory processes with opposing effects, leading to differential cellular responses to subsequent stimuli.

Moreover, through a closer investigation of the delayed negative feedback, we found that cell cycle phases at IFN treatment onset differentially influence USP18 upregulation, resulting in heterogeneous delays in the induction of this signaling suppressor. More specifically, the G1 and early S phases enable an open window for immediate USP18 upregulation upon the IFN treatment. If cells are exposed to IFN when they pass the G1/S phases, the IFN treatment cannot initiate USP18 induction until the G1 phase of the next cell cycle, causing a dramatic delay in the induction kinetics. In this way, individual cells exhibit substantial cell-to-cell variability in their USP18 induction delay times, as observed experimentally (*Figure 4D*), which arises from their different cell cycle phases at IFN treatment onset. Delays in the expression of key regulatory factors have been widely observed in signal-dependent transcriptional responses and have been found important for ensuring proper execution of biological functions (*Hoffmann et al., 2002*; *Covert et al., 2005*; *Lemmon et al., 2016*). Mechanisms underlying these delays include extensive nucleosome occupancy at the promoter regions (*Hansen and O'Shea, 2015*; *Sen et al., 2020*) and cascades in gene expression programs (*Killip et al., 2014*). Gene regulatory networks with specific topologies can also give rise to delayed gene expression, enabling filtering of transient inputs (*Shen-Orr et al., 2002*; *Mangan et al., 2003*; *Toettcher et al., 2013*; *Thurley et al., 2018*). The cell cycle gating revealed in this study represents a new mechanism that leads to delayed induction of specific genes upon stimulation. Distinct from other mechanisms, in this scenario, the cell cycle serves as a control hub that connects the expression of key pathway regulators with extracellular and intracellular conditions, enabling potential cross-regulation of IFN signaling by various factors. For example, external serum or nutrient conditions, which influence cell cycle progression, could thereby modulate USP18 expression and impact IFN responsiveness. This mode of regulation may also provide a link between cell cycle stages and the effects of IFNs on cancer cells. For instance, the resistance of cancer stem cells to IFN treatments may be attributed partially to the fact that they spend the majority of time in G0/G1 (*Doherty et al., 2017*), resulting in high USP18 expression and low IFN responsiveness. Along the same line, a recent report showed that IFN in combination with an G2/M inhibitor increased necroptosis in cancer cells (*Frank et al., 2019*), possibly due to delayed USP18 induction.

We acknowledge that our study only considered the control of specific ISGs, *STAT1*, *IRF9*, and *USP18*, which play crucial regulatory roles in IFN-driven signaling. There are other mechanisms that also contribute to priming or desensitization of the JAK-STAT pathway. For instance, it has been recently shown that acquisition of histone H3 methylation upon IFN pretreatment accelerates the recruitment of RNA polymerase II and transcription factors, leading to primed transcriptional responses to re-stimulation (*Kamada et al., 2018*). For desensitization, the family of SOCS proteins represses JAK-STAT signaling at multiple layers via various mechanisms (*Tamiya et al., 2011*), constituting negative feedback loops in addition to that mediated by USP18. The coupled ISGF3-USP18 feedback system revealed in this study may function together with all the other mechanisms to maintain homeostasis in the responses to varying IFN signals. Further work will be needed to evaluate the relative contributions of different mechanisms and how they coordinate to fine-tune the IFN responsiveness.

Future studies will also be needed to determine the detailed mechanisms underlying the cell cycle-gated regulation of USP18. Our results suggest that DNA methylation at the promoter region may be involved in mediating the effects of cell cycle on gene expression (*Figure 5D*). However, more experiments will be required to directly monitor DNA methylation level over time and measure its effects on chromatin modifications and the recruitments of transcription factors and RNA polymerase II. Intriguingly, ISGs have a wide range of CpG site numbers, and potentially methylation levels, in their promoters. Among the 278 ISGs examined, 114 genes, besides USP18, have more than 100 CpG sites in their promoters (*Figure 5—figure supplement 2, A*). It would be interesting to perform a genome-wide analysis to see whether the cell-cycle-gated regulation can be more generally observed in other ISGs and how this regulatory scheme contributes to the innate immune response at a global level. Furthermore, IFN-α induces proinflammatory and antiviral responses in many types of cells. As our current study has focused only on HeLa cells, it is necessary to extend the study to other non-cancerous epithelial cell lines. In addition, it would be interesting for future studies to examine and compare the dynamics and regulation of IFN-driven signaling and gene expression responses in other cell types, such as macrophages, dendritic cells, natural killer cells, and T cells, all of which are important players in mounting innate and adaptive immune responses. With specialized physiologies and functions, these cells may exhibit largely distinct response dynamics and modes of regulation. A careful investigation along this direction will advance our understanding about how various types of cells utilize the same set of molecular pathways with different kinetics to communicate with one another and coordinate their responses to infection. In addition, our study, and many recent single-cell analyses from other groups (*Jeknić et al., 2019*), highlight the presence of substantial clonal heterogeneity in cellular responses to signals. An open question that deserves extensive further exploration is how these cell-to-cell variabilities contribute to biological functions, in vitro, and more importantly, in vivo, under physiological contexts. Ongoing and future technological advances will enable us to tackle the emerging questions and challenges in single-cell biology, providing a more comprehensive, quantitative, and dynamic view of biological systems.

Finally, due to the COVID-19 pandemic caused by SARS-CoV-2, there is an urgency in understanding the basic biology of host response to viral infection, which may help the development of clinical strategies against the disease. Latest research revealed that SARS-CoV-2 is especially sensitive to type I IFNs, compared to other coronaviruses, making IFN pretreatment a potential strategy to prevent SARS-CoV-2 infection (*Lokugamage et al., 2020*; *Blanco-Melo et al., 2020*; *Sallard et al., 2020*). Our findings suggested possible ways to enhance the effectiveness of IFNs for future clinical use. For example, repetitive short-term administrations of IFN may lead to less desensitization and a more dramatic effect than prolonged treatments. Alternatively, the combined use of USP18 inhibitors with type I IFNs may substantially boost the effectiveness of IFNs against SARS-CoV-2. Further studies in animal models will be needed to test these strategies and to direct clinical applications for treating the disease.

# Materials and methods

## Key resources table

| Reagent type (species) or resource | Designation | Source or reference | Identifiers | Additional information |
|---|---|---|---|---|
| Cell line (*Homo sapiens*) | HeLa $P_{ACTB}$-NLS-2iRFP-P2A-ACTB | This paper | NHM003 | Available upon request from Hao lab |
| Cell line (*Homo sapiens*) | HeLa $P_{ACTB}$-NLS-2iRFP-P2A-ACTB, $P_{STAT1}$-STAT1-mCherry | This paper | NHM008 | Available upon request from Hao lab |
| Cell line (*Homo sapiens*) | HeLa $P_{ACTB}$-NLS-2iRFP-P2A-ACTB, $P_{STAT1}$-STAT1-mCherry, $P_{IRF9}$-IRF9-P2A-mCitrine | This paper | NHM025 | Available upon request from Hao lab |
| Cell line (*Homo sapiens*) | HeLa $P_{ACTB}$-NLS-2iRFP-P2A-ACTB, $P_{STAT1}$-STAT1-mCherry, $P_{IRF9}$-IRF9-P2A-mCitrine, shRNA USP18 | This paper | NHM026 | Available upon request from Hao lab |

*Continued on next page*

*Continued*

| Reagent type (species) or resource | Designation | Source or reference | Identifiers | Additional information |
|---|---|---|---|---|
| Cell line (*Homo sapiens*) | HeLa $P_{ACTB}$-NLS-2iRFP-P2A-ACTB, $P_{STAT1}$-STAT1-mCherry, $P_{IRF9}$-IRF9-P2A-mCitrine, shRNA USP18 negative control | This paper | NHM027 | Available upon request from Hao lab |
| Cell line (*Homo sapiens*) | HeLa $P_{ACTB}$-NLS-2iRFP-P2A-ACTB, $P_{STAT1}$-STAT1-mCherry, $P_{IRF9}$-IRF9-P2A-mCitrine, shRNA SOCS1 | This paper | NHM031 | Available upon request from Hao lab |
| Cell line (*Homo sapiens*) | HeLa $P_{ACTB}$-NLS-2iRFP-P2A-ACTB, $P_{STAT1}$-STAT1-mCherry, $P_{IRF9}$-IRF9-P2A-mCitrine, $P_{USP18}$-NLS-mCerulean-USP18 | This paper | NHM032 | Available upon request from Hao lab |
| Cell line (*Homo sapiens*) | HeLa $P_{ACTB}$-NLS-2iRFP-P2A-ACTB, $P_{USP18}$-NLS-mCerulean-USP18, $P_{CMV}$-DHB-mCherry | This paper | NHM035 | Available upon request from Hao lab |
| Cell line (*Homo sapiens*) | HeLa $P_{ACTB}$-NLS-2iRFP-P2A-ACTB, $P_{USP18}$-NLS-mCerulean-USP18, $P_{CMV}$-mCh-Gem1-P2A-CFP-Cdt1 | This paper | NHM036 | Available upon request from Hao lab |

## Cell cultures

HeLa and HEK293T were ordered from ATCC (ATCC CCL-2 and ATCC CRL-11268, respectively) and tested negative for *Mycoplasma* using *Mycoplasma* detection kit (Southern Biotech). HeLa and HEK293T cells were cultured in Dullbecco minimal essential medium (DMEM: Thermo Scientific HyClone #SH30022FS) supplemented with 10% fetal bovine serum, 4 mM L-glutamine, 100 I.U./ml penicillin and 100 µg/ml streptomycin at 37˚C, 5% $CO_2$ and 90% humidity. The imaging media is phenol red free DMEM (Life Technology-Gibco) with identical supplements as the culture medium. The transfections were performed with 1 µg DNA: 2 µl Fugene HD (Promega E2311) ratio. Cells were seeded at 300,000 cells/well in six-well plate for 18 hrs before transfection. Two days after transfection puromycin was added to the medium at 1 µg/ml, and cells were selected for 2 days. Survival cells were grown for another 7 days before sorted with FACS into 96-well and expanded into monoclonal cell lines.

## Drug treatments

Cells were treated with 100 ng/mL recombinant human IFN-α 1a (Prospec: cyt-520). Lovastatin (Selleck Chemicals, S2061) was used at 5 µM and Roscovitine (Sigma-Aldrich, R7772) was used at 5 µM. Decitabine (Sigma-Aldrich, A3656) was used at 100 µM. We note that we did not observe any cell death in the decitabine experiments. In our system, decitabine at 100 µM slowed down cell growth, but did not cause apoptosis at least during the time scales of our experiments. However, we did observe cell death at 200 µM decitabine.

## Cell line construction

We followed the CRISPR/Cas9 protocol (*Ran et al., 2013*) to construct the reporter cell line. In general, the gRNAs were designed by online CRISPR tool (http://crispr.mit.edu) and the DNA oligos were ordered from Eurofins Genomics, annealed and cloned into pSpCas9(BB)−2A-Puro (Addgene #48139) vector plasmids. gRNA plasmids were transfected into HEK293T cells and tested for gRNA efficiency using the T7 endonuclease assay. Only the most efficient gRNA was used with the donor DNA. The donor plasmids were constructed using Gibson assembly method. We used site-directed in-vitro mutagenesis to make synonymous substitution in the donor plasmids to avoid gRNA recognition and Cas9 cutting of the linearized donor DNA.

In more detail, we first developed a nuclear marker cell line by inserting the nuclear localization signal followed by two copies of infrared fluorescent protein (NLS-2xiRFP) (*Ogrodnik et al., 2014*) under the endogenous actin promoter followed by a P2A spacer in HeLa cells. This cell line ensured a constitutive expression without introducing exogenous strong constitutive promoter and greatly assisted cell segmentation and tracking. Briefly, the gRNA and the linearized donor DNA were

transfected into HeLa cells and the transfected cells were screened with 1 µg/ml puromycin for 2 days. The cells were allowed to grow for additional 5 days before sorted by FACS. The fluorescently positive cells were sorted as single cell into 96-well plate. We collected at least 500 single cells. We grew the cells for additional 3 weeks to obtain homogenous clones. On average, about 30% of cells formed colonies and all were screened for fluorescent signal with the microscope. A minimal of 10 clones were then genotyped and checked for homozygosity and correct integration using at least three pairs of primers and confirmed with sequencing. Positive clones were further validated with western blot to ensure correct protein expression. After construction and validation, the engineered single-clonal cell line was assigned a unique identification number, entered in our electronic database, and stored in liquid nitrogen with a cryoprotectant. The same procedure was performed for CRISPR-based tagging the additional genes, *STAT1*, *IRF9* and *USP18* sequentially.

The knockdown of *USP18* by shRNA was done using retrovirus transduction. We screened the transfected cells with 1 µg/ml puromycin for 5 days and confirmed the presence of the construct in the cells with PCR and confirmed the knock-down of USP18 with western blotting. Similarly, the knockdown of *SOCS1* by shRNA was done using lentivirus transduction, screened and validated with PCR and qPCR.

For the cell cycle reporter cell lines, we transfected *USP18* CRISPR constructs into the nuclear marker cell line and screened for a correct and homogenous monoclonal clone. We then used lentivirus to stably integrate pCMV-DHB-mCherry or pCMV-mCherry-Geminin(1-110)-P2A-mCitrine-Cdt1 (30-120) in pLenti-Puro (Addgene: 39481). Cells were screened with puromycin and sorted by FACS to generate monoclonal cell lines.

Primers used in this study were listed in *Supplementary file 1*. Plasmids and cell lines constructed in this study were listed in *Supplementary file 2* and Key Resources Table, respectively.

## Microfluidic and cell culture setup for time-lapse microscopy

Fabrication of the microfluidic device was conducted as described previously (*Kolnik et al., 2012*). For setting up the microfluidic experiments, HeLa cells were washed with dPBS and detached from the culture dish with 0.25% trypsin EDTA, centrifuged at 200 rcf for 3 min and resuspended with the complete imaging medium at a density of 7–10 million cells per mL. The suspension was loaded into the microfluidic device and allow the cells to adhere for at least 36 hrs in the standard incubator (37°C, 5% $CO_2$ and 90% humidity). The detail of the loading protocol is described previously (*Kolnik et al., 2012*). The device was set up in a customized chamber with 5% $CO_2$ and 37°C. The flow of the media was 1 ml/hr and the control of the valves were done with customized Arduino board.

For experiment performed on 24-well tissue culture plate, cells were seeded at 25,000 cells/well for 18 hrs before the treatments. Cells were washed with PBS and replaced with new medium before setting with the microscope to acquire images.

## Image acquisition

Time-lapse images were acquired using a Nikon Ti-E inverted microscope equipped with integrated Perfect-Focus (PFS), Nikon Plan Apochromat Lambda objective lens, and Evolve 512 EMCCD camera (Teledyne Photometrics). Time-lapse imaging was performed with an on-stage incubator equipped with temperature control and humidified 5% $CO_2$. Images were taken every 5 min for phase, Cy5.5 and mCherry channel and every 20 min for YFP and 30 min for CFP using Nikon NES element software.

## Image analysis and single-cell tracking

Background correction was performed using ImageJ 'rolling ball' background subtraction algorithm with 50-pixel radius. Nuclear segmentation was done using the nuclear marker iRFP reporter and then refined by marker-based watershed and the mask was generated. The phase images were used to generate masks for whole-cell segmentation. The masks were applied to other channels to quantify fluorescent intensity. Single-cell segmentation, tracking and quantification were performed using a custom MATLAB code developed in our lab, as described previously (*Li et al., 2017a*; *Li et al., 2017b*).

## Cell cycle phase inference

From the single-cell time traces, we used nuclear and cell morphology changes as markers to identify each cell division, since HeLa cells become rounded and non-adherent when dividing. For CDK2 activity reporter, within one cell cycle G1, S and G2 were identified based on the intensity and dynamics of nuclear DHB-mCherry. Similarly, for the FUCCI reporter, G1 phase was classified as the phase between the end of cell division and the time the YFP-Cdt1 signal reaches maximum. S phase began as YFP-Cdt1 signal starts to decline until it intersects with mCherry-Gem1 signal. The rest of the cell cycle was considered the G2 phase.

## Viral replication assay

Cells were pretreated with 100 ng/ml IFN-$\alpha$ for 0, 10 or 24 hrs and washed with PBS three times followed by 8 hrs of normal medium. Cells were infected by adding normal medium containing 2500 plaque forming units (PFU) of vesicular stomatitis virus (VSV), which corresponds to a multiplicity of infection (MOI) of ~0.01. Two hours after infection, medium was replaced again with normal medium. Viral supernatant was collected 18 hrs post-infection. Viral titer was quantified by plaque assay on BHK cells.

## Immunoblotting

Immunoblotting was performed as previously described in detail (*Arimoto et al., 2017*).

## Computational modeling

### Deterministic model

The simplified kinetic model of the gene regulatory network consists of two species, IRF9 and USP18, and they impose positive and negative regulation to gene expression, respectively. The positive regulation by IRF9 is represented by the function $pf$:

$$pf = k_1 \cdot \frac{IRF9}{k_2 + IRF9}$$

and the negative regulation by USP18 is represented by the function $nf$:

$$nf = \frac{k_3}{k_3 + USP18}$$

The expression of IRF9 and USP18 are both regulated by interferon (IFN) input and a combination of these two functions, and are governed by the ordinary differential equations (ODEs) below:

$$\frac{d}{dt} IRF9 = I(t) \cdot (k_4 + pf) \cdot nf$$

$$\frac{d}{dt} USP18 = I(t) \cdot S_u \cdot (k_5 + pf) \cdot nf$$

where I(t) is the IFN input, taking either 0 or 1. $S_u$ is a stepwise function that generates a delay in USP18 upregulation:

$$S_u = \begin{cases} 0, \text{ when the IFN input time} < \tau \\ 1, \text{when the IFN input time} \geq \tau \end{cases}$$

where $\tau$ is the delay time of USP18 production.

During the time scale of our experiments, we observed very modest decays of STAT1, IRF9 and USP18. Therefore, to keep our model simple, we considered only protein production, but not decay. We used the model to quantitatively analyze the USP18 delay time. To this end, we systematically assigned the value of $\tau$ to be from 1 to 20 hours, while keeping all the other parameters free. We fit the model to the data for each assigned value of $\tau$. The ODEs were solved using custom MATLAB code based on the basic Euler's method with dt=0.001. Fitting was done using MATLAB built-in function, *lsqcurvefit*. The time trace data of P$_{IRF9}$-YFP reporter fluorescence under sustained IFN-$\alpha$ stimulation (*Figure 1C*) and the data from all the pretreatment experiments (*Figure 1F*) were used

for the fitting. We found that the fitting error (between simulations and data) reaches the minimum when $\tau$ is 8 hours (*Figure 3B and C*). We included the fitting results for each delay time in *Figure 3—figure supplement 1B*. As shown in the figure, shorter delay times resulted in less desensitization effects from the 24-hr pretreatment, whereas longer delay times led to slightly less priming effects from the 2-hr pretreatment, compared with the data. We note that, these fitting analyses supported the hypothesis that there should be a substantial delay in USP18 induction to account for the observed effects, but did not provide sufficient power to quantitatively constrain the value of the delay time. The best-fit parameters are summarized in *Supplementary file 3*. The simulated time trace using the best-fit parameters was shown in *Figure 3—figure supplement 1C*. We note that our model, due to its simplicity, has limited capability to reproduce the dynamic response data. Therefore, we weighted more on the endpoint data under different conditions than the time trace data during fitting. The sensitivity analysis all the parameters was included in *Figure 3—figure supplement 1D*. To evaluate the roles of positive feedback and negative feedback loops on model performance, we compared the performance of three models: (1) the original model with both positive and negative feedback loops, (2) the model without positive feedback loop; and (3) the model without negative feedback loop. To this end, we computed Akaike information criterion (AIC) using the best-fit parameters for each model. The distribution of the data was assumed to be Gaussian, and the likelihood function was computed as:

$$L = \prod \frac{1}{\sqrt{2\pi\sigma^2}} e^{\frac{-(y-\tilde{y})^2}{2\sigma^2}}$$

where y is experimental result and $\tilde{y}$ is the predicted result. Akaike information criterion is computed as:

$$AIC = 2k - 2\ln(L)$$

where k is the number of parameters and L is the computed likelihood function. We found that the original model performs better than the other two models with the smallest AIC, suggesting that both positive and negative feedback loops are important for the model to reproduce experimental data (*Figure 3—figure supplement 1E*).

We further used the model to predict the responses to repetitive versus sustained IFN inputs. In particular, we simulated the responses to a $5 \times 8$-hr input and a 40-hr sustained input. We found that the $5 \times 8$-hr input induced a higher expression of IRF9 than that induced by the 40-hr input, and this difference is USP18-dependent (*Figure 3*, D and E). This prediction was validated by experiments, as shown in *Figure 3F*.

All the code in this study is stored at https://github.com/yaj030/2020_Sorn_Elife (*Jiang, 2020*; copy archived at https://github.com/elifesciences-publications/2020_Sorn_Elife).

## Incorporation of cell cycle regulation in the model

As described in the main text, based on the experimental results in *Figures 4* and *5*, we assumed that USP18 expression can be induced immediately if the IFN input starts within an open window of a cell cycle and the delay time would be 0. However, if the input misses the window, the cell will have to wait until the open window of the next cell cycle and the delay time will equal to the waiting time. The length of a full cell cycle was estimated to be 21.82 hrs based on the experimental data (*Figure 2—figure supplement 1*). The model was fit to the experimentally-determined distribution of USP18 delay times within a cell cycle (*Figure 6B*; the data from *Figure 4E*) to obtain an estimate of the open window length as 7.3 hrs. The stochastic simulation was performed with the stochastic differential equations:

$$\frac{d}{dt}IRF9 = I(t) \cdot (k_4 + pf) \cdot nf + \xi_{IRF9}$$

$$\frac{d}{dt}USP18 = I(t) \cdot S_u(k_5 + pf) \cdot nf + \xi_{USP18}$$

in which, $\xi_{IRF9}$ and $\xi_{USP18}$ are white noise terms, representing expression noise for the two species. Both of the two terms are Gaussian random variables with a mean of zero and the standard

deviations set to be 250 and 1000, respectively. These values of noise terms were manually chosen from observing the single-cell data in *Figure 6D*. The input onset times of single cells within a cell cycle are random numbers uniformly distributed within a cell cycle between 0 to 21.82 hours. $S_u$ is no longer a deterministic stepwise function with a fixed delay $\tau$, instead it is a stochastic stepwise function, where the delay $\tau$ varies depending on the IFN treatment onset within a cell cycle, as described above. Other parameters are the same as those listed in *Supplementary file 3*. We simulated the single-cell responses of the pretreatment experiments with different input durations. For each pretreatment condition, we performed 400 simulations where each simulation represented a single cell exposed to the IFN inputs. Our simulated results were plotted as scatter plots in *Figure 6C*.

To compare the simulation results with data, we calculated the correlation coefficients of *IRF9* and *USP18* in *Figure 6C* (simulation) and 6D (data), and obtained −0.37 and −0.49, respectively. In addition, we calculated the coefficients of variance (CVs) of single cells for *IRF9* and *USP18* under different pretreatment conditions. For *IRF9*, we obtained 0.44, 0.52, and 0.46 upon 2-hr, 10-hr, and 24-hr pretreatments, respectively, from the simulated results. We obtained 0.28, 0.37, and 0.56 upon 2-hr, 10-hr, and 24-hr pretreatments, respectively, from the data. For *USP18*, we obtained 0.71, 0.59, and 0.24 upon 2-hr, 10-hr, and 24-hr pretreatments, respectively, from the simulated results. We obtained 0.41, 0.49, and 0.35 upon 2-hr, 10-hr, and 24-hr pretreatments, respectively, from the data. Based on these analyses, we conclude that our stochastic simulations can qualitatively reproduce the negative correlation between USP18 expression upon pre-treatment and IRF9 induction by the second stimulation, but cannot quantitatively reproduce the expression levels or the cell-to-cell variance under different conditions.

## Acknowledgements

We thank Dr. Daniel Kaganovich (University Medical Center Gottingen, Germany) for generously providing the plasmids for NLS-iRFPx2 plasmid and Dr. Sabrina Spencer (University of Colorado Boulder) for the DHB-mVenus plasmid. This work was supported by NIH R01 GM111458 (to NH), NIH R01 CA177305 (to D-EZ), NIH R01 CA232147 (to D-EZ), NIH R35 GM133633 and Pew Biomedical Research scholars (to MDD), the DPST training scholarship from the Royal Thai government (to AM), and a T32 training grant GM007240 (to APR).

## Additional information

### Funding

| Funder | Grant reference number | Author |
| --- | --- | --- |
| National Institute of General Medical Sciences | GM111458 | Nan Hao |
| National Cancer Institute | CA177305 | Dong-Er Zhang |
| National Cancer Institute | CA232147 | Dong-Er Zhang |
| National Institute of General Medical Sciences | GM133633 | Matthew D Daugherty |
| Pew Charitable Trusts | Pew Biomedical Research Scholars | Matthew D Daugherty |
| Government of Thailand | DPST training scholarship | Anusorn Mudla |
| National Institute of General Medical Sciences | GM007240 Training Grant | Andy P Ryan |

The funders had no role in study design, data collection and interpretation, or the decision to submit the work for publication.

### Author contributions

Anusorn Mudla, Conceptualization, Formal analysis, Investigation, Methodology, Writing - original draft, Writing - review and editing; Yanfei Jiang, Bingxian Xu, Formal analysis, Investigation,

Methodology, Writing - original draft, Writing - review and editing; Kei-ichiro Arimoto, Matthew D Daugherty, Formal analysis, Investigation, Methodology, Writing - review and editing; Adarsh Rajesh, Andy P Ryan, Wei Wang, Formal analysis, Investigation, Methodology; Dong-Er Zhang, Conceptualization, Formal analysis, Investigation, Methodology, Writing - review and editing; Nan Hao, Conceptualization, Resources, Formal analysis, Supervision, Funding acquisition, Investigation, Methodology, Writing - original draft, Project administration, Writing - review and editing

## Author ORCIDs
Anusorn Mudla (iD) https://orcid.org/0000-0003-1132-2307
Yanfei Jiang (iD) http://orcid.org/0000-0002-4479-1874
Matthew D Daugherty (iD) http://orcid.org/0000-0002-4879-9603
Nan Hao (iD) https://orcid.org/0000-0003-2857-4789

## Decision letter and Author response
Decision letter https://doi.org/10.7554/eLife.58825.sa1
Author response https://doi.org/10.7554/eLife.58825.sa2

## Additional files

### Supplementary files
- Supplementary file 1. Primers used in this study.
- Supplementary file 2. Plasmids constructed in this study.
- Supplementary file 3. Best-fit parameter values of the model.
- Transparent reporting form

### Data availability
All data generated or analysed during this study are included in the manuscript and supporting files.

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
