## [Decision Letter]

**Acceptance summary:**

In this paper, the authors used a combination of experimental analyses and mathematical modeling to discover opposing effects of IFN (priming vs desensitization) in one cellular system. The results suggest that a delayed up-regulation of a negative feedback protein (*USP18*) can explain the opposite functional effect of varying input durations of IFN. By studying heterogeneous behaviors of single cells, the authors revealed a cell cycle-dependent, promoter methylation-mediated mechanism underlying the expression of this protein. Discovery of these IFN opposing effects in one system is novel, and reconciling the paradox by different kinetics of positive and negative feedback loops is fundamentally important for understanding complex IFN signaling. This study shows how a combination of detailed experimental analyses and mathematical modeling can reveal fundamental insights on individual cellular signaling dynamics.

**Decision letter after peer review:**

Thank you for submitting your article "Cell cycle-gated feedback control mediates desensitization to interferon stimulation" for consideration by *eLife*. Your article has been reviewed by three peer reviewers, one of whom is a member of our Board of Reviewing Editors, and the evaluation has been overseen by Aleksandra Walczak as the Senior Editor. The reviewers have opted to remain anonymous.

The reviewers have discussed the reviews with one another and the Reviewing Editor has drafted this decision to help you prepare a revised submission.

As the editors have judged that your manuscript is of interest, but as described below that additional analyses are required before it is published, we would like to draw your attention to changes in our revision policy that we have made in response to COVID-19 (https://elifesciences.org/articles/57162). First, because many researchers have temporarily lost access to the labs, we will give authors as much time as they need to submit revised manuscripts. We are also offering, if you choose, to post the manuscript to bioRxiv (if it is not already there) along with this decision letter and a formal designation that the manuscript is "in revision at *eLife*". Please let us know if you would like to pursue this option. (If your work is more suitable for medRxiv, you will need to post the preprint yourself, as the mechanisms for us to do so are still in development.)

Summary:

In this paper “Cell cycle-gated feedback control mediates desensitization to interferon stimulation”, the authors used endogenous labeling and time-lapse microscopy, and discovered that IFN leads to opposing effects (priming vs desensitization), depending on durations of its pretreatment. Modeling and experimental results suggest a delayed up-regulation of negative feedback protein *USP18* can explain the opposite function of different input duration. Further, the authors studied heterogeneous behaviors of single cells and revealed a cell cycle-dependent, promoter methylation-mediated mechanism underlying the expression of *USP18*.

Overall, this is an interesting study and the authors investigate a timely and relevant question. Discovery of IFN opposing effects in one system is novel, and reconciling the paradox by different kinetics of positive and negative feedback loops is fundamentally important for understanding complex IFN signaling. The manuscript is well written and the work is clearly presented. However, a more detailed and thorough analysis and discussion of the presented data and the used mathematical model is suggested to support the proposed mechanisms. In particular this applies to the following points:

a) The key conclusions of the manuscript indicating the influence of cell cycle phases on regulation of *USP18* expression and heterogeneity should be corroborated by additional analyses and discussion of alternative hypotheses.

b) The statistical analyses and representation, especially for the single-cell data, has to be improved.

c) The analyses of the mathematical model could be improved by making an effort for more quantitative comparisons to data, indicating uncertainty in parameter estimates and by relating it to previous analytical work.

These aspects will be subject to re-review to determine if these issues have been satisfactorily addressed. We would like to point you to the specific points raised below that contain some suggestions of how to improve the analyses.

Essential revisions:

1) A key conclusion of this manuscript is that cell cycle phases regulate *USP18* expression. Figure 5A is the essential and sole evidence to support this causality: G1/S arrest by synchronization results in earlier *USP18* up-regulation, and thus shorter delay time. However, without showing the distribution of activation time for *USP18* and IRF9, it is not clear whether this is the case. An alternative explanation of the shorter delay time would be delayed IRF9 expression by synchronization. Or, some combination of the two possibilities. Re-analysis of the imaging data may be able to address this.

2) The authors proposed that differential cell cycle phases is the reason for cell-to-cell variation of *USP18* expression. However, Figure 5C suggests cell cycle only partially correlates with *USP18* expression (~half G2 cells have the same activation time as G1/S cells). Alternatively, or in addition, expression/activity of the positive regulator ISGF3 may contribute to the heterogeneous *USP18* expression. Indeed, Figure 4—figure supplement 1 shows variations of nuclear/cyto STAT1 ratio at single cell level. The degree to which pre-existing heterogeneity in ISGF3 contributes to the observed heterogeneity in *USP18* expression should be tested. A reasonable step could be to reanalyze the relationship of *USP18* expression with STAT1 expression/activation or IRF9 expression.

3) A differential regulation of the positive regulator IRF9 and negative regulator *USP18* by promoter methylation was proposed. Direct evidence is necessary to support this claim, e.g. show the distributions of both IRF9 and *USP18* activation time in control and decitabine.

4) Kinetic model (Figure 3): to keep the model simple, the authors considered only protein production but not decay. However, based on this assumption, it would seem that repetitive IFN pulses (Figure 3D) should lead to IRF9 accumulation after each round of pulse and ultimately activation of *USP18*, just as in the sustained pretreatment case. The authors need to support this critical modeling assumption.

5) In the adjusted model (Figure 6), the authors assume that the IFN input onset time is uniformly distributed within a cell cycle. However, this is inconsistent with the fact that HeLa cells spend the longest time in G1, moderate time in S, and progress fast through G2/M.

6) The authors collected an impressive amount of kinetic single-cell data, giving new insight into the cellular mechanisms of the interferon response. Unfortunately, that data is in large parts simply plotted, and not analyzed systematically. Therefore, it is not always clear how to interpret the extra information from single-cell data. In many cases only the average is taken from several independent single-cell experiments (e.g. Figure 1F, Figure 3F). In other cases, properties such as "bimodality" (Figure 4D) or "different kinetics" (Figure 4C) are just stated heuristically without any quantitative analysis. Even though a stochastic mathematical model is included (Figure 6), no effort is made to quantitatively compare model results and single-cell data. In Figure 6C, stochastic simulations of the model are able to qualitatively reproduce the observed experimental data. Although measured in arbitrary units, the quantitative measurements for the IRF9 induction seem to be smaller than in the experimental data (~roughly only half the average values) especially for short periods of IFN-pretreatment, with the 2h-simulations also not matching the trend observed in the experimental data between the different conditions. How sensitive are these predictions dependent on the parameter choices for the model? Given the complexity of the model and the type of data used for fitting, parameter identifiability might be impaired.

7) At various instances, the manuscript falls short on statistical analysis. It is not always clear how error bars are calculated, and whether or not all shown single-cell data points stem from the same experiment or the same set of experiments. Further, image quantification and statistical tests are almost entirely missing.

8) The presented mathematical model is a valuable addition to the paper, with the potential to both rationalize and quantify the experimental results. However, it should be pointed out that the structure of the model follows a well-studied topology, namely the "persistence detector" motif (Shen-Orr et al., 2002; Mangan et al., 2003), which was more recently also studied in kinetic single-cell experiments (Toettcher, Weiner and Lim, 2013) and in a data-driven model of cell-cell communication (Thurley, Wu and Altschuler, 2018). Therefore, it is not surprising that the model captures an improved response to a pulsed input as compared to a sustained input, as shown in Figure 3. In the presented model, the original feed-forward loop is replaced by direct delay (as in some other implementations such as Thurley et al.), and is supplemented by additional, coupled positive and negative feedback loops. It remains unclear to what extent that loop is required for the observed effect, since analysis remains limited to variation of the delay time (Figure 3B). More specifically – would the simple delay mechanism alone, without any feedback, be sufficient to explain the data in Figure 3C? And how much do the individual feedbacks contribute to the final results?

---

## [Author Response]

Essential revisions:1) A key conclusion of this manuscript is that cell cycle phases regulate USP18 expression. Figure 5A is the essential and sole evidence to support this causality: G1/S arrest by synchronization results in earlier USP18 up-regulation, and thus shorter delay time. However, without showing the distribution of activation time for USP18 and IRF9, it is not clear whether this is the case. An alternative explanation of the shorter delay time would be delayed IRF9 expression by synchronization. Or, some combination of the two possibilities. Re-analysis of the imaging data may be able to address this.

We thank the reviewers for this insightful suggestion. We have now provided the distributions of activation times for both *IRF9* and *USP18* in new Figure 5—figure supplement 1. As shown in the figure, G1/S arrest leads to earlier *USP18* upregulation in cells treated with serum starvation, lovastatin, and roscovitine. In contrast, *IRF9* activation times were affected very modestly by either serum starvation or lovastatin. Roscovitine results in a slight delay in *IRF9* upregulation but a dramatic acceleration in *USP18* upregulation. Therefore, the shorter delay times in Figure 5A were mainly due to earlier *USP18* upregulation upon G1/S arrest. We have also included, in the figure legend, the mean, the 95% confidence interval, and coefficient of variance (CV) for each distribution to demonstrate these changes quantitatively. We have added text to describe these results and clarify this point in subsection “Cell cycle phases differentially regulate *USP18* expression”.

2) The authors proposed that differential cell cycle phases is the reason for cell-to-cell variation of USP18 expression. However, Figure 5C suggests cell cycle only partially correlates with USP18 expression (~half G2 cells have the same activation time as G1/S cells). Alternatively, or in addition, expression/activity of the positive regulator ISGF3 may contribute to the heterogeneous USP18 expression. Indeed, Figure 4—figure supplement 1 shows variations of nuclear/cyto STAT1 ratio at single cell level. The degree to which pre-existing heterogeneity in ISGF3 contributes to the observed heterogeneity in USP18 expression should be tested. A reasonable step could be to reanalyze the relationship of USP18 expression with STAT1 expression/activation or IRF9 expression.

We appreciate this thoughtful suggestion. As requested, we have analyzed the relationship of *USP18* expression with that of *IRF9* or STAT1 at the single-cell level and showed the scatter plots and correlation coefficients in new Figure 4—figure supplement 2. We observed that the heterogeneity in these upstream components contributes partially to cell-to-cell variation in the expression levels of USP18. However, the delay times of *USP18* upregulation correlate poorly with STAT1 or IRF9 expression. We have added text to describe and discuss these results in subsection “The kinetics of *USP18* upregulation by IFN is heterogeneous in single cells”.

3) A differential regulation of the positive regulator IRF9 and negative regulator USP18 by promoter methylation was proposed. Direct evidence is necessary to support this claim, e.g. show the distributions of both IRF9 and USP18 activation time in control and decitabine.

We thank the reviewers for this request. We have now provided the distributions of activation times for both *IRF9* and *USP18* in control and decitabine in new Figure 5—figure supplement 4. As shown in the figure, the decitabine treatment shortened the activation times for *USP18*, while modestly affecting *IRF9* expression. We have also included, in the figure legend, the mean, the 95% confidence interval, and coefficient of variance (CV) for each distribution to demonstrate these changes quantitatively. We have added text to describe these results in subsection “Cell cycle phases differentially regulate *USP18* expression”.

4) Kinetic model (Figure 3): to keep the model simple, the authors considered only protein production but not decay. However, based on this assumption, it would seem that repetitive IFN pulses (Figure 3D) should lead to IRF9 accumulation after each round of pulse and ultimately activation of USP18, just as in the sustained pretreatment case. The authors need to support this critical modeling assumption.

We thank the review for this suggestion. During the time scales of our experiments, we observed very modest decays of STAT1, IRF9 and USP18. We clarified this in Materials and methods subsection “Computational modeling”. In the model (Figure 3E), the short pulsatile input is less able to induce USP18 upregulation than the sustained input, because the expression delay of USP18 filters out the short inputs. This weaker induction of the negative regulator USP18 accounts for the higher IRF9 induction by the pulsatile input. We have modified the text in subsection “Computational modeling suggests a delayed negative feedback loop through *USP18*” to clarify this point.

5) In the adjusted model (Figure 6), the authors assume that the IFN input onset time is uniformly distributed within a cell cycle. However, this is inconsistent with the fact that HeLa cells spend the longest time in G1, moderate time in S, and progress fast through G2/M.

We apologize for the unclarity of our model description. In our simulations, we assumed that the IFN onset time for each single cell is a random time point uniformly distributed within a cell cycle. However, in our model, the periods of different cell cycle phases, e.g. G1, S and G2/M, are not equally distributed within a cell cycle, and hence the probabilities of the IFN input onset time falling at different cell cycle phases are not equal. We have revised the text in subsection “Cell cycle-gated feedback control shapes single-cell responses to repetitive IFN inputs” to clarify this assumption.

6) The authors collected an impressive amount of kinetic single-cell data, giving new insight into the cellular mechanisms of the interferon response. Unfortunately, that data is in large parts simply plotted, and not analyzed systematically. Therefore, it is not always clear how to interpret the extra information from single-cell data. In many cases only the average is taken from several independent single-cell experiments (e.g. Figure 1F, Figure 3F). In other cases, properties such as "bimodality" (Figure 4D) or "different kinetics" (Figure 4C) are just stated heuristically without any quantitative analysis.

We thank the reviewer for the positive comments. As for the request of additional analysis and quantification, we have now performed quantification and statistical analysis for the data in Figure 1F, Figure 2, Figure 4, Figure 5 and their figure supplements.

Even though a stochastic mathematical model is included (Figure 6), no effort is made to quantitatively compare model results and single-cell data. In Figure 6C, stochastic simulations of the model are able to qualitatively reproduce the observed experimental data. Although measured in arbitrary units, the quantitative measurements for the IRF9 induction seem to be smaller than in the experimental data (~roughly only half the average values) especially for short periods of IFN-pretreatment, with the 2h-simulations also not matching the trend observed in the experimental data between the different conditions. How sensitive are these predictions dependent on the parameter choices for the model? Given the complexity of the model and the type of data used for fitting, parameter identifiability might be impaired.

We thank the reviewer for the request of additional analyses of modeling results. We have now performed parameter sensitivity analysis and the quantitative comparison between model results and the data. We also added discussions about the limitations of the model.

7) At various instances, the manuscript falls short on statistical analysis. It is not always clear how error bars are calculated, and whether or not all shown single-cell data points stem from the same experiment or the same set of experiments. Further, image quantification and statistical tests are almost entirely missing.

We apologize for the unclarity of data presentation and the lack of statistical analysis and quantification. We have now added the missing information, quantification, and statistical tests.

8) The presented mathematical model is a valuable addition to the paper, with the potential to both rationalize and quantify the experimental results. However, it should be pointed out that the structure of the model follows a well-studied topology, namely the "persistence detector" motif (Shen-Orr SS et al., 2002; Mangan et al., 2003), which was more recently also studied in kinetic single-cell experiments (Toettcher, Weiner and Lim, 2013) and in a data-driven model of cell-cell communication (Thurley, Wu and Altschuler, 2018). Therefore, it is not surprising that the model captures an improved response to a pulsed input as compared to a sustained input, as shown in Figure 3. In the presented model, the original feed-forward loop is replaced by direct delay (as in some other implementations such as Thurley et al.), and is supplemented by additional, coupled positive and negative feedback loops. It remains unclear to what extent that loop is required for the observed effect, since analysis remains limited to variation of the delay time (Figure 3B). More specifically – would the simple delay mechanism alone, without any feedback, be sufficient to explain the data in Figure 3C? And how much do the individual feedbacks contribute to the final results?

We thank the reviewer for this insightful comment and for pointing us to relevant previous studies. We have now added text in paragraph two of the Discussion to further our discussion about network motifs and gene expression delay and cited these relevant papers.

In addition, we performed sensitivity analysis for all the parameters in the model. As shown in new Figure 3—figure supplement 1C, both positive and negative feedback loops (governed by k_1_, k_2_ and k_3_) are important for the observed effects in Figure 3C. For example, if the strength of positive feedback loop is too weak (k_1_ is too small), the priming effect by 10-hr pretreatment will be abolished. Similarly, if the activation threshold for the negative feedback is too high (k_3_ is too large), the desensitization effect by 24-hr pretreatment will be abolished. We have added text in subsection “Computational modeling suggests a delayed negative feedback loop through *USP18*” to describe these results.